# Construction of a depression risk prediction model for hepatitis B patients based on machine learning strategy

**Siyi Wang**[1], **Haoqi Liu**[1], **Chen Liang**[2], **Chui Kong**[3], **Jingchun Li**[1], **Min Wang**[1], **Kaiqiang Dong**[1], **Qianqi Wang**[1], **Dong Zhang**[1], **Rongjuan Guo**[1]*

1 The Second Clinical Medical College, Beijing University of Traditional Chinese Medicine, Beijing, China, 2 School of Acupuncture-Moxibustion and Tuina, Beijing University of Traditional Chinese Medicine, Beijing, China, 3 School of Information Science and Technology, Fudan University, Shanghai, China

* dfguorongjuan@163.com

## Abstract

### Background

Hepatitis B (HBV) is a chronic viral infection that can lead to cirrhosis, liver failure, and liver cancer, and has a profound impact on the patient's mental health. However, current depression screening mainly relies on self-filled scales and clinical experience, lacking objective and efficient prediction tools. This study aims to construct a risk prediction model for depression in hepatitis B patients based on machine learning, and explore the key features that affect the occurrence of depression, so as to optimize mental health management strategies.

### Methods

This study used the NHANES database to collect demographic, dietary, physical examination, laboratory test and questionnaire data. The data were standardized and SMOTE oversampling was used to solve the problem of class imbalance. Random Forest (RF) was used for feature screening to identify the top 20 most important predictive features, and five machine learning models (Gradient Boosting, Logistic Regression, AdaBoost, MLPClassifier, LDA) were used for prediction. The model performance was evaluated by AUC (area under the curve), accuracy, recall, precision and F1-score, and ROC curves, calibration curves, and decision curve analysis (DCA) were drawn to evaluate the clinical applicability of the model.

### Results

All five machine learning models performed well in the task of predicting the risk of depression in hepatitis B patients, among which MLPClassifier (multi-layer perceptron) performed best, with an AUC of 0.935, a recall of 0.980, and an F1-score of 0.917, which was better than other models. In addition, feature analysis results

**Data availability statement:** All the NHANES data are available online (https://wwwn.cdc.gov/nchs/nhanes/search/), and the original data of the studied subjects has been uploaded as a supporting information file entitled "Manuscrpt availability data.zip.

**Funding:** This work was supported by 1. the National Natural Science Foundation of China (Grant No. U21A200276). 2. Beijing Major Difficult Diseases Collaborative Research Project of Traditional Chinese and Western Medicine ( 2023BJSZDYNJBXTGG-014 ).

**Competing interests:** The authors have declared that no competing interests exist.

showed that liver function damage (serum total bilirubin, alkaline phosphatase), electrolyte imbalance (serum potassium ions), chronic inflammation (red blood cell distribution width, lymphocyte count), and socioeconomic factors (poverty-income ratio, race) were important factors affecting the risk of depression in hepatitis B patients.

## Conclusion

This study constructed an efficient and objective machine learning model that can be used to predict the risk of depression in patients with hepatitis B, providing a new tool for accurate screening and individualized management. The study revealed the potential mechanisms of physiological, biochemical and socioeconomic factors in the occurrence of depression in patients with hepatitis B, and provided a reference for future mental health intervention strategies.

---

## 1 Introduction

Depression is a prevalent psychiatric disorder, characterized by significant and persistent anhedonia (loss of pleasure) and lack of interest. It is associated with a high suicide rates, widespread prevalence, substantial disability, and significant recurrence [1]. Depression has become a major public health concern, affecting over 300 million individuals globally [2]. According to the World Health Organization(WHO), depression is currently the second leading cause of disability worldwide and is projected to become the primary contributor to the global disease burden by 2030 [3,4].

Hepatitis B (HBV) is a viral liver disease primarily transmitted through blood, bodily fluids, and mother-to-child transmission. The disease can present as an acute flare or progress to a chronic condition. Chronic hepatitis B may lead to severe liver damage, including cirrhosis, hepatic failure, and liver cancer. A recent study estimated the global prevalence of hepatitis B at 3.2%, equating to approximately 257.5 million individuals [5], thus posing a substantial challenge to global public health. China, in particular, has a high prevalence of HBV, with 86 million carriers, including 32 million individuals with chronic hepatitis B [6].

The rates of depression and anxiety in hepatitis B patients are significantly higher than those in the general population. This heightened incidence may be attributed to the long-term psychosocial stress these patients experience [7]. Depression in HBV patients not only impacts the progression of the disease but is also strongly linked to adverse mortality outcomes [8]. Studies have indicated a positive correlation between depression and liver-related mortality [9], with a stronger association observed in HBsAg-positive individuals [10]. linked to depression [11,12].

The pathophysiology of depression involves neuroinflammatory responses, where immune reactions within the central nervous system (e.g., microglial activation and the release of pro-inflammatory cytokines) induce oxidative stress [13]. Several studies suggest that both HBV and depression share common biological mechanisms, such as immune system dysregulation and neuroinflammation [14]. Psychological interventions tailored to hepatitis B patients have been shown to positively influence

disease progression and overall prognosis [15,16]. Therefore, early identification of depression in HBV patients is critical. However, the diagnosis of depressive disorders remains largely dependent on the clinical experience of physicians and subjective assessments using specific scales [17], as no objective biomarkers currently exist for depression. This highlights the urgent need for objective diagnostic tools or technological means to assist in diagnosis.

Machine learning (ML) is a widely used artificial intelligence (AI) technique, has been extensively applied in various biomedical fields for the automated analysis of complex data. ML offers significant advantages in interpreting omics data, identifying biomarkers, and constructing predictive models for precision medicine [18–20].

With the rapid advancement of AI technology, research focused on machine learning for depression prediction and diagnosis has grown substantially. Prior studies have employed machine learning to develop models predicting blood biomarkers for depression [20,21] and to construct predictive models based on neuroimaging biomarkers in patients with depression [22,23]. Additionally, research has utilized machine learning to analyze smartphone behavioral patterns in depression patients [24], further demonstrating its potential in clinical practice. However, to date, no predictive model exists for the risk of depression in hepatitis B patients, representing a critical gap in the field.

In this study, we retrieved cohort studies from the NHANES database, which includes populations with hepatitis B and PHQ-9 assessments. Feature data were standardized using the StandardScaler, and data imbalance was addressed using the Synthetic Minority Over-sampling Technique (SMOTE). Feature selection was performed using the Random Forest algorithm, with the top 20 most important features identified. These features were then used to train five machine learning models— GradientBoost, Logistic Regression, Adaboost, MLPClassifier, and Linear Discriminant Analysis (LDA)—for predictive model construction. The workflow of this study is shown in Fig 1.

The process includes data acquisition from NHANES (2009–2020), preprocessing (cleaning, standardization, SMOTE oversampling), feature selection using Random Forest, training of five machine learning models, and evaluation via five-fold cross-validation. The final output is a depression risk stratification (high vs. low risk).

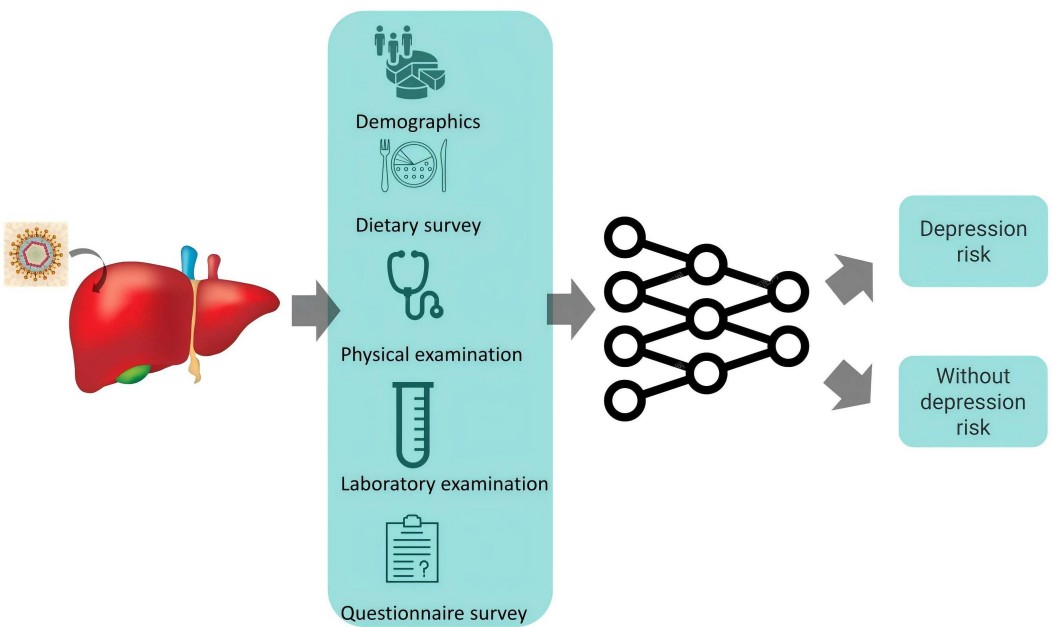

**Fig 1. Workflow of the study.**

## 2 Method

### 2.1 Data acquisition

The data for this study came from the NHANES (National Health and Nutrition Examination Survey) database, which is a national, representative health and nutrition survey widely used in epidemiological studies. We identified 189 hepatitis B (HBV) patients with complete serological profiling (anti-HBc, HBsAg, anti-HBs) from NHANES 2009–2020, integrating demographic characteristics, dietary records, physical examinations, laboratory parameters, and psychosocial questionnaires to construct machine learning models.

Demographics Data covers basic socioeconomic characteristics of patients, such as age, gender, marital status, and educational attainment. These variables may affect the occurrence of depression, such as race, socioeconomic status, and access to medical care. Dietary Data records the patient's dietary intake, including macronutrients and micronutrients. Studies have shown that dietary patterns are closely related to mental health. For example, deficiencies in nutrients such as B vitamins, iron, and zinc may increase the risk of depression. Examination Data covers basic physiological indicators such as height, weight, blood pressure, and heart rate. These variables can be used to assess the patient's overall health and further analyze the relationship between physiological health and depression. Laboratory data include liver function (ALT, AST, bilirubin), blood lipid levels, inflammatory factors (CRP), and blood routine tests. These laboratory data provide biological evidence to help us explore potential biomarkers of depression risk. Questionnaire Data includes the PHQ-9 (Patient Health Questionnaire-9) depression screening questionnaire. PHQ-9 is a depression assessment tool currently widely used in clinical and research settings. It consists of 9 questions, each with a score range of 0–3 and a total score range of 0–27. The higher the score, the more severe the patient's depressive symptoms. This study defined depression status as a binary variable (0 or 1) based on the PHQ-9 score, with a total score ≥10 being considered as possible moderate or above depression.

During the data collection process, we strictly followed the NHANES data standards to ensure data integrity and exclude obvious outliers to improve the reliability of the model. At the same time, we used a unique patient identifier (SEQN) to integrate data of various categories to build a complete patient data set, providing high-quality input data for subsequent machine learning model training.

### 2.2 Data preprocessing and feature screening

In the data preprocessing stage, we first cleaned the data, removed obvious outliers, and filled in missing data. For numerical variables, the mean was used for filling, while categorical variables were filled with the mode to retain sample information to the greatest extent. In addition, we applied StandardScaler to normalize all continuous variables to ensure that different features have the same scale and reduce the impact of feature value range differences on model training.

Since the number of depression-positive samples of hepatitis B (HBV) patients in the NHANES dataset is relatively small, the data distribution is unbalanced. To solve this problem, we used the SMOTE (Synthetic Minority Over-sampling Technique) method to perform synthetic oversampling on the training data. SMOTE generates new samples in the feature space to make the distribution of minority class data more balanced, thereby improving the stability and generalization ability of the model in the depression prediction task.

During the feature screening process, we used the Random Forest (RF) algorithm to evaluate the importance of each feature and selected the top 20 variables that contributed most to the prediction of depression. These variables cover demographic, laboratory test, and physical examination data, and can provide comprehensive predictive information at the physiological, immune, and psychological levels. Finally, the selected features were used for model training to improve the efficiency and interpretability of the model.

## 2.3 Machine learning prediction model construction

In the process of building the machine learning prediction model, we used five models: Gradient Boosting (GB), Logistic Regression (LR), AdaBoost, MLPClassifier (Multi-layer Perceptron) and Linear Discriminant Analysis (LDA), and optimized their parameters to ensure the best prediction effect.

For parameter settings, we performed preliminary grid search and manual tuning for each model. We performed L2 regularization (penalty = 'l2') on Logistic Regression (LR), selected solver=' lbfgs ' as the optimization solver, and set the maximum number of iterations to max_iter = 1000 to ensure convergence. For Gradient Boosting (GB), we set n_estimators = 300, learning rate learning_rate = 0.05, and maximum depth max_depth = 4 to control model complexity and prevent overfitting. AdaBoost uses Decision Tree as the base learner, and its maximum weak classifier depth is set to base_estimator = DecisionTreeClassifier (max_depth = 2), learning rate learning_rate = 0.1, and the number of weak classifiers n_estimators = 200. MLPClassifier (Multilayer Perceptron) uses a three-layer neural network structure with a hidden layer size of hidden_layer_sizes = (128, 64, 32), activation function activation = ' relu ', optimizer solver = ' adam ', L2 regularization parameter alpha = 0.0001, maximum number of iterations max_iter = 500 to ensure sufficient training. LDA (Linear Discriminant Analysis) uses default parameters for classification to provide baseline performance in the linearly separable case. Throughout the training process, we strictly used the training set for model fitting and the validation set for performance evaluation, and no data leakage occurred.

## 2.4 Model evaluation

In the model evaluation stage, we used accuracy, recall, precision, F1-score and area under the curve (AUC) as the main evaluation indicators to measure the classification performance of the model in depression risk prediction. We introduced 5-fold cross-validation to calculate the mean AUC (CV AUC Mean) and its standard deviation (CV AUC Std) to evaluate the stability and generalization ability of the model. To further analyze the performance of the model, we drew ROC curves to show the classification ability of each model, calibration curves to evaluate whether the probability output of the model matches the actual classification results, and decision curve analysis (DCA) to measure the clinical benefits of the model at different decision thresholds.

## 3 Results

### 3.1 Sociodemographic characteristics

This study included 189 patients diagnosed with hepatitis B. The demographic characteristics are summarized as follows: the mean age was 49.03 ± 16.12 years; 107 patients (56.61%) were male and 82 (43.39%) were female. An analysis of marital status, based on an effective sample of 183 participants (missing data: 6), revealed that 105 (57.38%) were married, 22 (12.02%) were widowed, 20 (10.93%) were divorced, 5 (2.73%) were separated, and 30 (16.39%) had never been married. Regarding educational attainment, based on an effective sample of 182 participants (missing data: 7), 86 patients (47.25%) had less than postsecondary education, while 96 (52.75%) had attained postsecondary education or higher.

### 3.2 Experimental setup

In this study, we extracted a dataset of hepatitis B (HBV) patients from the NHANES database and conducted experiments according to five-fold cross validation. Specifically, all HBV patient data included in the analysis were divided into five subsets. In each iteration, four of these subsets (approximately 80%) were used as training sets for model training and parameter optimization, while the remaining subset (approximately 20%) was used as a validation set to evaluate model performance. This process was repeated five times, ensuring that every sample participated in both training and validation, thereby comprehensively assessing the model's stability and generalization capabilities. 5-fold cross-validation not only improves efficiency with limited sample size but also effectively mitigates potential bias that might arise from a single partitioning of the model. During model evaluation, we aggregated performance metrics (such as AUC and

F1-score) across all folds and averaged them as the final result. The hyperparameters of all machine learning models were optimized to ensure optimal performance. The computing environment of this study included the Ubuntu 20.04 operating system, Python 3.8 as the programming language, and core libraries including scikit-learn, imbalanced-learn, numpy, and pandas. The experiments were run on a server with an NVIDIA RTX 3090 GPU (24GB video memory) to ensure efficient model calculation and optimization. The training of all models converged, and learning rate adjustment and regularization strategies were used to ensure that there was no overfitting.

### 3.3 Feature screening results

Feature importance was ranked using the Random Forest algorithm, identifying the top 20 predictive features for depression risk in this HBV cohort (Fig 2). These key features encompassed laboratory indicators and socioeconomic factors: liver function markers (serum total bilirubin – LBXSTB, alkaline phosphatase – LBXSAPSI); electrolytes (serum potassium – LBXSKSI, serum calcium – LBDSCASI, LBXSCA); hematological/inflammatory markers (hemoglobin measures

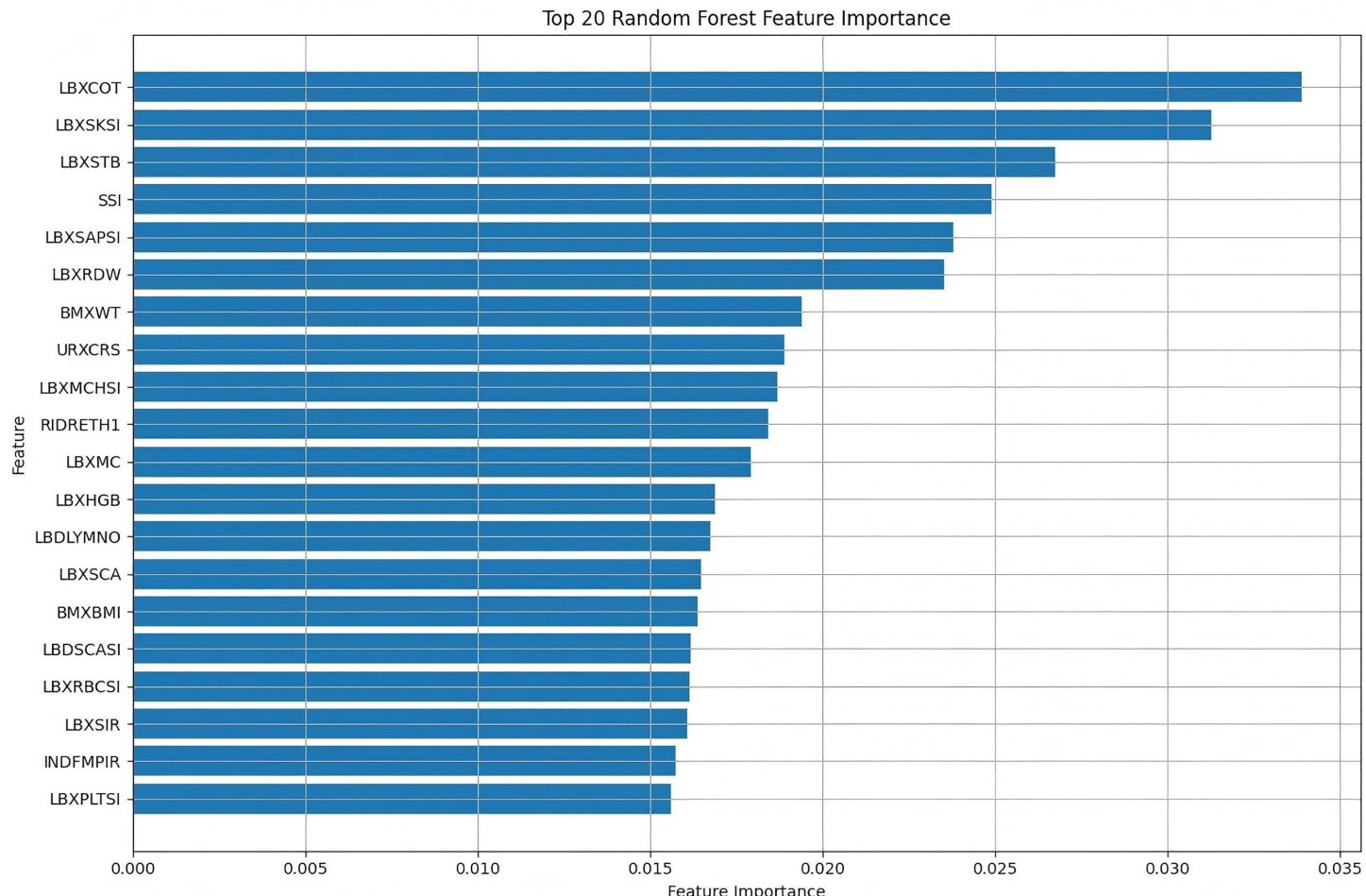

**Fig 2. Top 20 important features selected by random forest algorithm.** Features are ordered by importance score (mean decrease in Gini impurity). Key categories include liver function markers (LBXSTB, LBXSAPSI), electrolytes (LBXSKSI, LBXSCA, LBDSCASI), hematological/inflammatory indices (LBXHGB, LBXMC, LBXMCHSI, LBXRBCSI, LBXRDW, lymphocyte count, platelet count), and socioeconomic factors (RIDRETH1, INDFMPIR). Abbreviations follow NHANES variable naming conventions.

– LBXMCHSI, LBXMC, LBXHGB; red blood cell count – LBXRBCSI; red blood cell distribution width – LBXRDW; lymphocyte count; platelet count – PLT); and socioeconomic factors (race/ethnicity - RIDRETH1; poverty income ratio – INDFMPIR).

### 3.4  Risk prediction model results

This study used five machine learning models, including Logistic Regression (LR), AdaBoost, MLPClassifier (multi-layer perceptron), Linear Discriminant Analysis (LDA) and Gradient Boosting (GB), to predict the risk of depression in hepatitis B patients, and evaluated the model performance based on indicators such as Accuracy, Recall, Precision, F1-score and AUC (area under the curve).

The experimental results show (Table 1) that the MLPClassifier model performs best, with an AUC of 0.935, a recall rate of 0.980, and an F1 value of 0.917, indicating that the model has high sensitivity and stability in predicting the risk of depression in hepatitis B patients. This means that the MLPClassifier has a strong risk identification ability and can effectively detect high-risk patients. Secondly, Gradient Boosting achieved an AUC of 0.919 and an F1 value of 0.851. The model performed well in balancing precision and recall. The AUC of AdaBoost also reached 0.881, proving that it has a strong ability to fit the data, but it is slightly inferior to MLPClassifier in terms of recall and overall prediction stability. In contrast, Logistic Regression and LDA performed poorly in all indicators, with AUCs of 0.739 and 0.751, respectively, indicating that linear models have certain limitations when dealing with complex feature spaces. Although Logistic Regression may provide better interpretability in some cases, its predictive ability is obviously inferior to other methods in the nonlinear feature environment of this study.

At the same time, we plotted ROC curves to compare the classification capabilities of each model (Fig 3), calibration curves to evaluate the consistency between the model's predicted probability and the actual classification (Fig 4), and decision curve analysis (DCA) to evaluate the clinical decision-making benefits of the model at different thresholds (Fig 5). The experimental results show that MLPClassifier performed best in all evaluation indicators, indicating that the model can more accurately capture high- risk individuals and avoid missed diagnoses. Gradient Boosting, as another preferred model, performs well in accuracy and stability, and is suitable for application scenarios that require a balance between precision and recall. In contrast, Logistic Regression and LDA have relatively low predictive capabilities due to their weak ability to fit the nonlinear features of the data.

## 4  Discussion

This study utilized the NHANES database to develop and rigorously evaluate multiple machine learning models for predicting depression risk in patients with hepatitis B virus (HBV) infection. Our predictive framework, incorporating data standardization, SMOTE oversampling, Random Forest feature selection, and optimized model training, demonstrated robust performance. The Multi-layer Perceptron classifier (MLP Classifier) demonstrated superior predictive capability, achieving an area under the receiver operating characteristic curve (AUC) of 0.935 and an exceptionally high recall of 0.980. This highlights the efficacy of deep learning architectures in capturing complex, nonlinear interactions within multimodal clinical

**Table 1.  Performance comparison of each model in the validation set.**

| Accuracy | Recall | Precision | F1 | AUC | Model |
|---|---|---|---|---|---|
| 0.6759 | 0.7255 | 0.6379 | 0.6789 | 0.7399 | LogisticRegression |
| 0.8056 | 0.8824 | 0.7500 | 0.8108 | 0.8813 | AdaBoostClassifier |
| 0.9167 | 0.9804 | 0.8621 | 0.9174 | 0.9353 | MLPClassifier |
| 0.6852 | 0.7255 | 0.6491 | 0.6852 | 0.7516 | LinearDiscriminantAnalysis |
| 0.8611 | 0.8431 | 0.8600 | 0.8515 | 0.9192 | GradientBoostingClassifier |

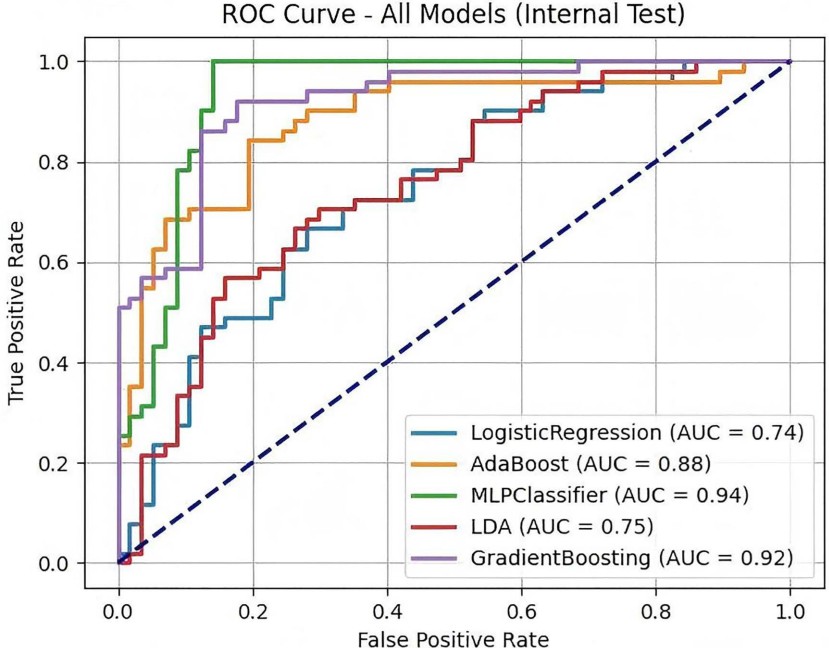

**Fig 3. ROC curves of each model and corresponding AUC values.** The MLPClassifier achieved the highest AUC (0.935), followed by Gradient Boosting (0.919), AdaBoost (0.881), Linear Discriminant Analysis (0.751), and Logistic Regression (0.739). Dashed diagonal line represents random classification.

data encompassing physiological, biochemical, and socioeconomic domains. Gradient Boosting (AUC = 0.919) also exhibited strong performance, leveraging its inherent feature importance assessment and ensemble structure. Conversely, the comparatively lower performance of Logistic Regression (AUC = 0.739) and Linear Discriminant Analysis (AUC = 0.751) underscores the likely nonlinear relationship between the identified risk factors and depression onset in this specific patient population.

Crucially, feature analysis identified the top 20 predictors via random forest, yielding significant biological and psychosocial insights. The prominence of liver function markers, specifically serum total bilirubin (LBXSTB) and alkaline phosphatase (LBXSAPSI), aligns with the established association between hepatic dysfunction and depression. Chronic HBV infection induces persistent hepatic and systemic inflammation [25], characterized by elevated pro-inflammatory cytokines exerting neurotoxic effects on emotion-regulating brain regions, thereby contributing to depression pathogenesis [26]. Dysregulated liver function may further directly impair neurotransmitter metabolism and exacerbate neuroinflammation. Features associated with hematopoiesis and inflammation—including hemoglobin levels (LBXHGB, LBXMC, LBXMCHSI), red blood cell indices (LBXRBCSI, LBXRDW), lymphocyte count, and platelet count (PLT)—likely reflect consequences of impaired hepatic synthetic function (e.g., reduced erythropoietin effectiveness, impaired vitamin B12/ folate synthesis) and potential antiviral therapy-induced bone marrow suppression [27] Anemia and associated fatigue, a core depressive symptom captured by the PHQ-9, may directly contribute to elevated depression scores. The significance of serum electrolytes, particularly potassium (LBXSKSI) and calcium (LBXSCA, LBDSCASI), underscores their fundamental roles in neuronal excitability and synaptic transmission. Dysregulation of potassium channels is increasingly implicated in mood disorders [28,29], while calcium ions are pivotal for neurotransmitter release and neuronal signaling. Key socioeconomic predictors included race/ethnicity

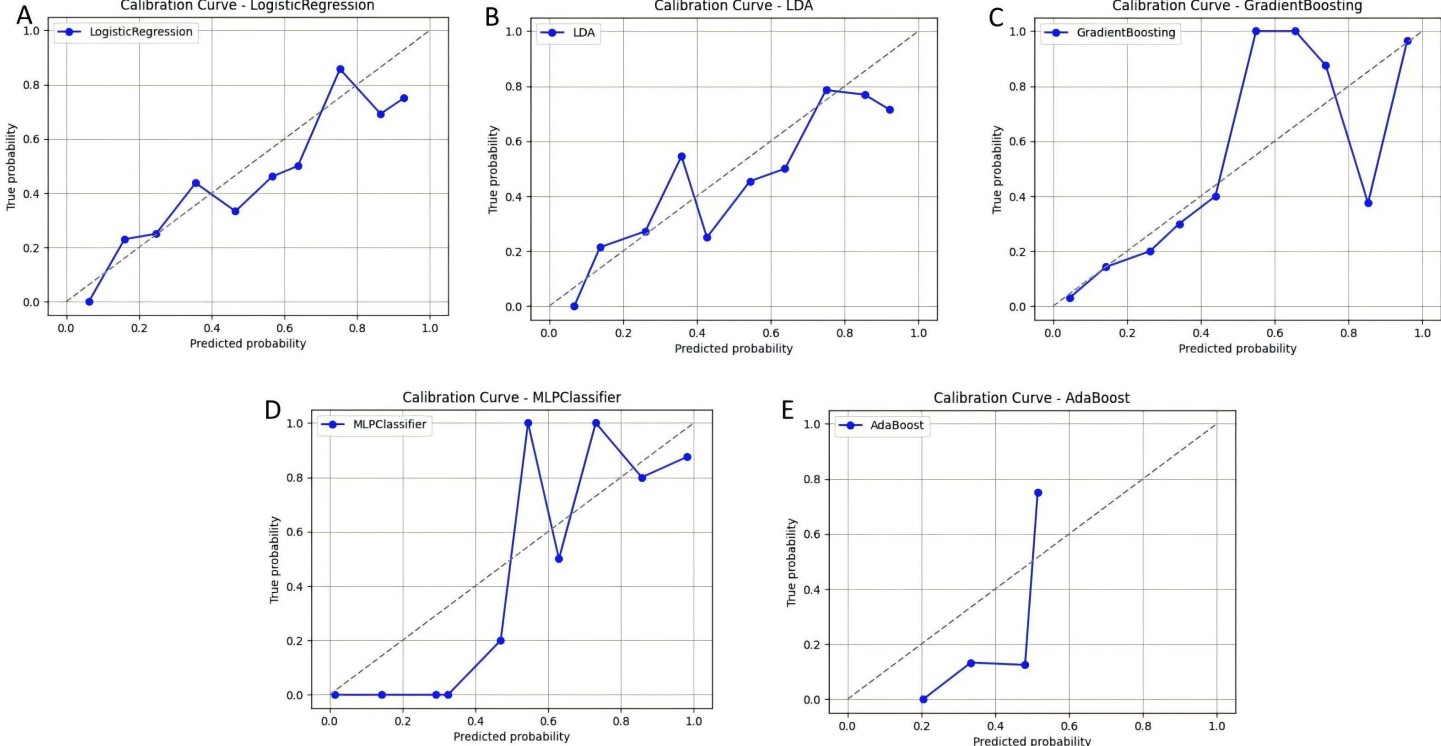

**Fig 4. Calibration curves of each model, where (AE) are the results of LR, LDA, GB, MLP and AdaBoost models respectively.** Plots show the relationship between predicted probability of depression risk (x-axis) and observed frequency (y-axis) for (A) Logistic Regression, (B) Linear Discriminant Analysis, (C) Gradient Boosting, (D) MLPClassifier, and (E) AdaBoost. The dashed diagonal line represents perfect calibration.

(RIDRETH1) and the poverty-income ratio (INDFMPIR), reflecting the substantial psychosocial burden, potential stigma [30], economic stress, and barriers to healthcare access faced by HBV patients, all recognized risk factors for depression [31,32]. This convergence of features underscores a multifactorial etiology wherein HBV-related physiological disturbances and treatment sequelae intersect with significant psychosocial stressors.

Integrating these diverse risk factors into a machine learning (ML)-based prediction model offers a potential paradigm shift from traditional screening reliant solely on subjective instruments like the PHQ-9. While valuable, PHQ-9 assessments are susceptible to patient self-reporting bias and mood fluctuations, potentially missing subclinical or latent cases. Our objective, data-driven approach, synthesizing routinely collected clinical and demographic data, could facilitate earlier identification of high-risk HBV patients within standard liver disease management protocols. Integration into Electronic Health Record (EHR) systems could enable automated risk stratification. Early identification could enable timely proactive interventions, including enhanced psychological monitoring, targeted counseling, or pharmacotherapy, with the potential to mitigate the detrimental impact of depression on HBV disease progression, treatment adherence, and overall quality of life [15,16].

However, several critical limitations warrant careful consideration and temper the interpretation of our findings. First, concerning model performance and potential overfitting: The exceptionally high recall (0.980) and AUC (0.935) observed for the MLPClassifier on the internal test set must be interpreted cautiously given the relatively small sample size (n = 189 HBV patients) and the application of SMOTE oversampling. Although SMOTE mitigates class imbalance, and regularization techniques were employed (e.g., L2 in Logistic Regression, alpha in MLPClassifier) while controlling model complexity

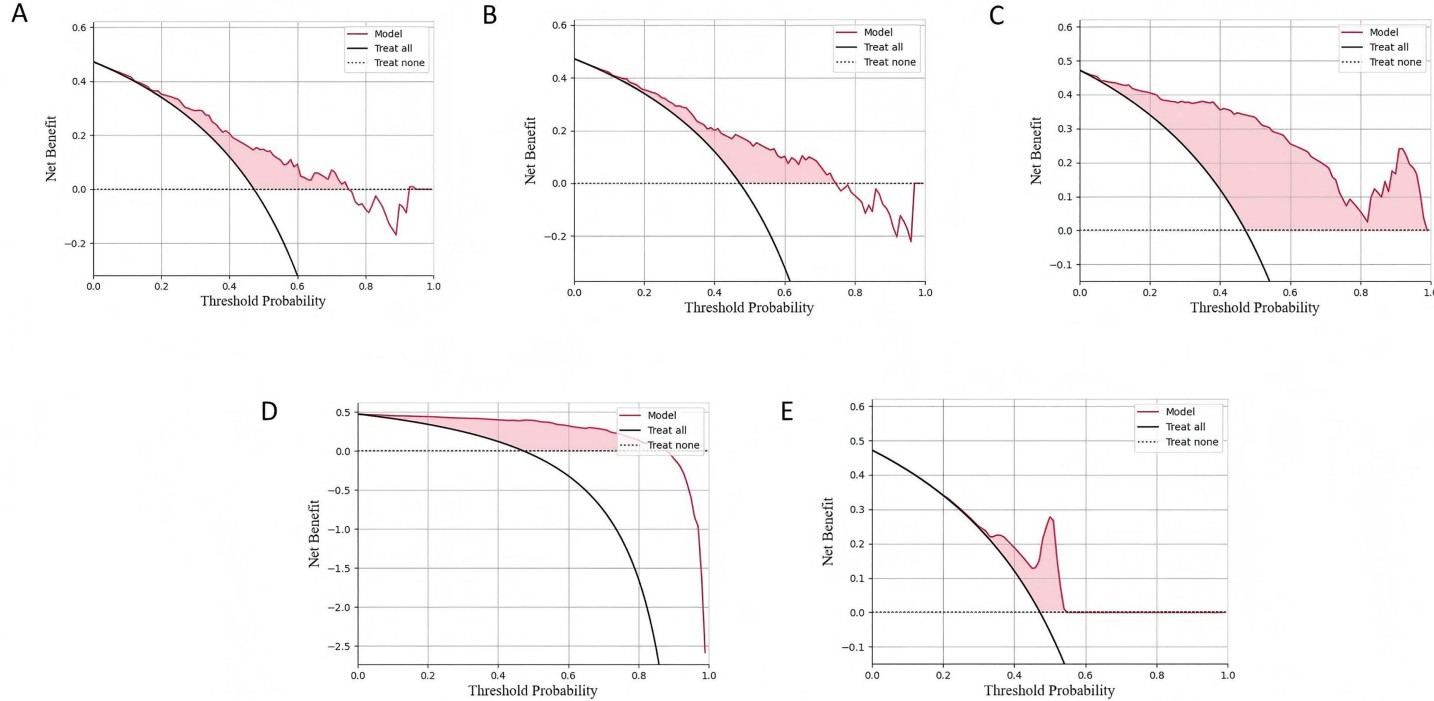

**Fig 5. DCA curves of each model, where (AE) are the results of LR, LDA, GB, MLP and AdaBoost models respectively.** Curves depict the net benefit of using each model for clinical decision-making compared to treating all patients (gray solid line) or treating none (gray dashed line). Models are shown for (A) Logistic Regression, (B) Linear Discriminant Analysis, (C) Gradient Boosting, (D) MLPClassifier, and (E) AdaBoost.

(e.g., max_depth in tree-based models), the risk of overfitting to idiosyncratic patterns within this limited NHANES cohort remains non-negligible. The high internal performance, exemplified by the 0.980 recall, may inflate estimates of the model's generalizability to real-world clinical settings. Second, The study has certain inherent limitations related to the dataset. While comprehensive, the National Health and Nutrition Examination Survey (NHANES) data are cross-sectional and retrospective in nature. Consequently, the analysis can only demonstrate associations between the identified risk factors and the onset of depression, though these findings may still provide some support for clinical decision-making. Furthermore, potential selection bias intrinsic to the survey methodology and the absence of detailed longitudinal psychiatric assessments beyond the PHQ-9 represent additional constraints on the analytical scope. Third, the lack of external validation constitutes a major limitation. Robust assessment of the model's clinical utility requires evaluation on independent, prospective cohorts from diverse clinical settings. Consequently, claims regarding clinical applicability must be considered preliminary. Future work should prioritize validation using larger, multi-center, prospective cohorts to confirm generalizability and refine performance estimates. Furthermore, while representative within NHANES, our feature set lacked granular psychological assessments, specific neurobiological markers (e.g., cytokines, neurotransmitter metabolites, neuroimaging data), genetic susceptibility factors, and detailed lifestyle/environmental data, which could enhance predictive accuracy and mechanistic insight. The limited interpretability ("black-box" nature) of the high-performing MLPClassifier also presents a barrier to clinical translation. Integrating explainable AI (XAI) techniques such as SHAP (Shapley Additive exPlanations) or LIME (Local Interpretable Model-agnostic Explanations) in future iterations is essential to elucidate feature-level contributions for individual predictions, thereby fostering clinician trust and enabling actionable insights.

In conclusion, this study successfully developed an ML framework demonstrating significant potential for predicting depression risk in HBV patients using readily accessible clinical and demographic data. The MLPClassifier model

exhibited outstanding internal performance, and the identified features provide valuable, multifaceted insights into the interplay of hepatic pathophysiology, neurobiological mechanisms, and socioeconomic adversity in depression development within this vulnerable population. For clinical translation, the model can be implemented as an automated screening tool within electronic health record systems, providing clinicians with an objective risk score to support patient stratification and management. By identifying high-risk individuals during routine follow-up visits, it facilitates timely mental health assessments and early interventions, thereby integrating depression risk monitoring into the standard care pathway for chronic hepatitis B. Despite these promising results, several limitations—including the small sample size, retrospective cross-sectional design, lack of external validation, and model interpretability challenges—warrant cautious interpretation and highlight necessary future directions. Subsequent studies should prioritize larger prospective cohorts, external validation, inclusion of multimodal data (e.g., omics and neurobiological markers), and development of interpretable models to translate this approach into clinically actionable tools for mental health management in chronic hepatitis B.

## 5  Conclusion

We develop and validate a machine learning framework predicting depression risk in hepatitis B patients, where a multilayer perceptron classifier achieves exceptional performance (AUC = 0.935, recall = 0.980). Through feature screening analysis, this study further revealed the impact of liver damage, electrolyte disorders, chronic inflammatory states, socioeconomic factors, etc. on the occurrence of depression in hepatitis B patients, providing a new direction for precision medicine and individualized intervention. The model of this study can be used to assist clinical decision-making, helping doctors identify high- risk patients earlier, optimize mental health management, and reduce the negative impact of depression on hepatitis B patients.

## Supporting information

**S1 Data.**
(ZIP)

## Author contributions

**Conceptualization:** Siyi Wang, Rongjuan Guo.

**Data curation:** Siyi Wang, Haoqi Liu, Chen Liang, Jingchun Li, Kaiqiang Dong, Dong Zhang.

**Formal analysis:** Kaiqiang Dong, Rongjuan Guo.

**Investigation:** Haoqi Liu, Chen Liang.

**Methodology:** Haoqi Liu, Rongjuan Guo.

**Project administration:** Siyi Wang, Min Wang, Kaiqiang Dong, Rongjuan Guo.

**Resources:** Siyi Wang, Chui Kong, Min Wang, Rongjuan Guo.

**Software:** Chui Kong, Min Wang, Qianqi Wang, Dong Zhang.

**Supervision:** Siyi Wang, Chui Kong, Rongjuan Guo.

**Validation:** Siyi Wang, Chui Kong, Jingchun Li, Min Wang, Qianqi Wang, Dong Zhang.

**Visualization:** Siyi Wang, Jingchun Li, Min Wang, Qianqi Wang.

**Writing – original draft:** Siyi Wang, Chen Liang.

**Writing – review & editing:** Siyi Wang, Chen Liang, Dong Zhang, Rongjuan Guo.

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
