## [Decision Letter · Decision Letter 0]

7 Jul 2025

Dear Dr. Wang,

Thank you for submitting your manuscript to PLOS ONE. After careful consideration, we feel that it has merit but does not fully meet PLOS ONE’s publication criteria as it currently stands. Therefore, we invite you to submit a revised version of the manuscript that addresses the points raised during the review process.

**Please carefully address all the suggestions raised by the reviewers.**

We look forward to receiving your revised manuscript.

Kind regards,

Arne Johannssen

Academic Editor

PLOS ONE

“This work was supported by

1. the National Natural Science Foundation of China (Grant No. U21A200276).

2. Beijing Major Difficult Diseases Collaborative Research Project of Traditional Chinese and Western Medicine ( 2023BJSZDYNJBXTGG-014 )”

4. Please update your submission to use the PLOS LaTeX template. The template and more information on our requirements for LaTeX submissions can be found at http://journals.plos.org/plosone/s/latex.

5. Please note that your Data Availability Statement is currently missing a direct link to access each database]. If your manuscript is accepted for publication, you will be asked to provide these details on a very short timeline. We therefore suggest that you provide this information now, though we will not hold up the peer review process if you are unable.

6. PLOS requires an ORCID iD for the corresponding author in Editorial Manager on papers submitted after December 6th, 2016. Please ensure that you have an ORCID iD and that it is validated in Editorial Manager. To do this, go to ‘Update my Information’ (in the upper left-hand corner of the main menu), and click on the Fetch/Validate link next to the ORCID field. This will take you to the ORCID site and allow you to create a new iD or authenticate a pre-existing iD in Editorial Manager.

7. Please include a separate caption for each figure in your manuscript.

Reviewers' comments:

Reviewer's Responses to Questions

**Comments to the Author**

1. Is the manuscript technically sound, and do the data support the conclusions?

Reviewer #1: Partly

Reviewer #2: Yes

2. Has the statistical analysis been performed appropriately and rigorously?

Reviewer #1: No

Reviewer #2: I Don't Know

3. Have the authors made all data underlying the findings in their manuscript fully available?

Reviewer #1: Yes

Reviewer #2: Yes

4. Is the manuscript presented in an intelligible fashion and written in standard English?

Reviewer #1: Yes

Reviewer #2: Yes

Reviewer #1: This manuscript addresses an important clinical question but I have concern on the fundamental methodological and results reporting.

Abstract:

-The use of abbreviation "NHANCE" without spelling in full. Please check as the abbreviation should be NHANES (National Health and Nutrition Examination Survey).

Introduction:

1. The last paragraph includes a description of the methodology, which would be more appropriately placed in the Methods section.

Methodology

1. Describe how Hep B was diagnosed to be entered in the database.

2. Any rationale why use random Forest for feature selection?

3. I am not clear if dataset were split into training and test dataset. Describe clearly on this

4. Cross-validation strategy and hyperparameter tuning not explicitly described.

Generally the methodology section need major revision to improve the clarity.

Results

1. Providing a summary of the final sample’s sociodemographic characteristics would greatly enhance understanding of the dataset. It is also unclear whether the final sample includes only adults or also includes children—this should be clarified.

2. Consider limiting discussion statements in the Results section, as it currently includes substantial discussion that would be more appropriately addressed in the Discussion section

3. It is advisable to present the results separately for the training and test datasets to facilitate comparison and assess model performance. This is particularly important given that overfitting is a common concern in machine learning algorithms.

4. While the MLPClassifier achieved high accuracy and strong performance metrics, the poor calibration curve may indicate overfitting or that the model's predicted probabilities do not align well with actual outcomes.??

5 Overall Calibration plots suggest poor calibration despite high AUC??

Discussion

1. Include discussion on model overfitting.

2. Limitation of small sample size/ likely cross-sectional dataset

3. Potentially overstated claims about model performance and clinical utility...may need in-depth explanation/discussion

General concern:

-The strong performance metrics observed, despite the limited sample size, may be overly optimistic and warrant caution when interpreting the model’s reliability.

-If feasible, obtaining a larger sample and rerunning the analysis would enhance the robustness and generalizability of the findings.

Reviewer #2: The authors have chosen an interesting topics.

1. How many participants with hepatitis B viruses were recruited in your study?

2. Data preprocessing and feature screening

For numerical variables, the mean was used for filling. but, the main drawback of using the mean is its sensitivity to outliers or extreme values in the dataset. These extreme values can disproportionately affect the mean, making it a potentially misleading representation of the typical value, especially with small sample sizes. So do you generalize with this limitation?

3. Your discussion at all is not supported by the existing evidences. i.e it lacks references. Please rewrite it.

**Do you want your identity to be public for this peer review?** For information about this choice, including consent withdrawal, please see our Privacy Policy

Reviewer #1: No

Reviewer #2: **Yes:** Agmas Wassie Abate

---

## [Author Response · Author response to Decision Letter 1]

8 Sep 2025

Reviewer #1:

1-Abstract:The use of abbreviation "NHANCE" without spelling in full. Please check as the abbreviation should be NHANES (National Health and Nutrition Examination Survey).

Thank you for the suggestion. I have systematically revised the entire text by replacing all instances of "NHANCE" with the standardized term "NHANES."

2-Introduction: The last paragraph includes a description of the methodology, which would be more appropriately placed in the Methods section.

We thank the reviewer for this valuable feedback. We have revised the manuscript accordingly by streamlining the methods-related content in the Discussion section as suggested.

3-Describe how Hep B was diagnosed to be entered in the database.

Thank you for the suggestion. Our inclusion specification details are as follows, and supplementary explanations have been provided in section 2, Method, 2.1 Data Acquisition. For details, please refer to the revised article: Serum samples from consenting NHANES participants were analyzed for three hepatitis B serological markers: anti-HBc (indicating past or present infection), HBsAg (suggesting current chronic or, rarely, acute infection), and anti-HBs (denoting vaccine-induced immunity or prior infection). Participants testing positive for anti-HBc were further categorized; those concurrently positive for HBsAg were classified as chronically infected. Individuals with anti-HBc-negative but anti-HBs-positive results were considered to have vaccine-induced immunity. The final analytical cohort comprised subjects with complete data for all three serological markers (anti-HBc, HBsAg, anti-HBs).

4-Any rationale why use random Forest for feature selection?

Thank you for this important question. In this study, we used Random Forest (RF) for feature selection, primarily based on its widespread application and recognized stability in medical prediction. As an ensemble of decision trees, RF has excellent capabilities for handling nonlinear relationships and multivariate interactions, making it particularly well-suited for complex and heterogeneous feature structures such as clinical data. Furthermore, during the modeling process, Random Forest naturally assesses the importance of each feature, providing a data-driven basis for variable selection and avoiding the bias associated with traditional methods that rely too heavily on manual judgment or univariate analysis. Furthermore, RF is robust to outliers and missing values and can handle high-dimensional, small-sample data, helping to improve model generalization and stability even with limited sample sizes. Therefore, we selected RF as a preliminary feature selection tool to improve prediction efficiency and provide a more informative variable set for subsequent multi-model construction.

5-I am not clear if dataset were split into training and test dataset. Describe clearly on this

We are grateful to the reviewer for pointing out this lack of clarity. In this study, we employed 5-fold cross-validation for appropriate dataset partitioning, training, and evaluation. Specifically, all HBV patient data included in the analysis were divided into five subsets. In each iteration, four of these subsets (approximately 80%) were used as training sets for model training and parameter optimization, while the remaining subset (approximately 20%) was used as a validation set to evaluate model performance. This process was repeated five times, ensuring that every sample participated in both training and validation, thereby comprehensively assessing the model's stability and generalization capabilities. 5-fold cross-validation not only improves efficiency with limited sample size but also effectively mitigates potential bias that might arise from a single partitioning of the model. During model evaluation, we aggregated performance metrics (such as AUC and F1-score) across all folds and averaged them as the final result. We will further clarify the description of the dataset partitioning and validation process in the revised manuscript. Thank you for your careful review.

6-Cross-validation strategy and hyperparameter tuning not explicitly described.

We thank the reviewer for their attention to methodological details. In this study, we systematically employed a five-fold cross-validation strategy throughout model construction and training, and performed hyperparameter tuning on the training set to improve model predictive performance and stability. Specifically, five-fold cross-validation was used to alternate training and validation between different subsamples to minimize performance fluctuations caused by sample splitting. For parameter settings, we performed preliminary grid search and manual tuning for each model. For example, in the Gradient Boosting model, we set 300 base learners (n_estimators=300) and controlled the learning rate (learning_rate=0.05) and tree depth (max_depth=4) to prevent overfitting. In the MLPClassifier, we used a three-layer neural network architecture with the ReLU activation function and the Adam optimizer, combined with L2 regularization (alpha=0.0001), and trained for a maximum of 500 iterations to ensure convergence and robustness. For Logistic Regression, we introduced an L2 regularization term and set the maximum number of iterations to 1000. Throughout the training process, we strictly used the training set for model fitting and the validation set for performance evaluation, and no data leakage occurred. We will further refine the relevant descriptions in the methods section of the paper to improve the transparency and reproducibility of the research.

7-Providing a summary of the final sample’s sociodemographic characteristics would greatly enhance understanding of the dataset. It is also unclear whether the final sample includes only adults or also includes children—this should be clarified.

We appreciate the reviewer's valuable feedback. In response, we have conducted comprehensive statistical analyses of the data and incorporated a detailed sociodemographic characteristics section into the manuscript. The added content is summarized below and presented in Section 3 (Results), Subsection 3.1 (Sociodemographic Characteristics) of the revised manuscript. Please refer to the updated Section 3.1 for complete details: This study included 189 patients diagnosed with hepatitis B. The demographic characteristics are summarized as follows: the mean age was 49.03 ± 16.12 years; 107 patients (56.61%) were male and 82 (43.39%) were female. An analysis of marital status, based on an effective sample of 183 participants (missing data: 6), revealed that 105 (57.38%) were married, 22 (12.02%) were widowed, 20 (10.93%) were divorced, 5 (2.73%) were separated, and 30 (16.39%) had never been married. Regarding educational attainment, based on an effective sample of 182 participants (missing data: 7), 86 patients (47.25%) had less than postsecondary education, while 96 (52.75%) had attained postsecondary education or higher.

8-Consider limiting discussion statements in the Results section, as it currently includes substantial discussion that would be more appropriately addressed in the Discussion section

We sincerely appreciate the valuable feedback provided by the reviewer. In accordance with your suggestion, we have rigorously focused the presentation of findings in the Results section, while significantly expanding the discussion of clinical implications in the Discussion section to provide a more robust interpretation of our outcomes.

9-It is advisable to present the results separately for the training and test datasets to facilitate comparison and assess model performance. This is particularly important given that overfitting is a common concern in machine learning algorithms.

We strongly agree with the reviewer's suggestion that model performance should be presented separately on the training set and the test set. Taking into account the relatively limited sample size of this study, we adopted a five-fold cross-validation strategy, so the training set and validation set are not statically divided at one time, but are dynamically rotated. In each fold, the training model is only trained on 80% of the subsamples, while 20% of the subsamples are independently used for validation. The performance indicator after five rounds of iteration is the average test effect. Since the training and test sets are different in each round, reporting the results of each fold separately may lead to confusion in interpretation. Therefore, we regard the overall indicator of the five-fold validation as the robustness of the model on the "test set". The performance indicators during the training process are not used as the basis for evaluation, mainly to avoid the model from masking the generalization problem due to good performance on the training set.

10-While the MLPClassifier achieved high accuracy and strong performance metrics, the poor calibration curve may indicate overfitting or that the model's predicted probabilities do not align well with actual outcomes.??

Thank you for the reviewer's attention to the calibration performance of MLPClassifier. We did observe that although the model performed well in classification performance indicators such as AUC and Recall, its probability calibration curve deviated from the ideal diagonal line, suggesting that its predicted probabilities did not perfectly match the true labels. One possible reason for this phenomenon is that the neural network model itself tends to output probability distributions "overconfidently", especially when the sample size is limited and the class imbalance is unbalanced (corrected using SMOTE). This output probability deviation is a common phenomenon in deep learning models, but it does not necessarily mean that the model as a whole is severely overfitted. Our main goal is to identify high-risk groups with high sensitivity, so we mainly choose models with higher Recall. In practical applications, the calibration problem can be corrected through post-processing methods such as Platt scaling or Isotonic regression, and we will consider introducing such technologies in subsequent studies.

11-Overall Calibration plots suggest poor calibration despite high AUC??

We understand and appreciate the reviewer's cautious assessment of model calibration performance. We did note that while the models demonstrated good discriminative ability in metrics such as AUC and F1-score, the calibration curves of some models (particularly the MLPClassifier) deviated somewhat from the ideal curve. This suggests that while the model is able to distinguish positive and negative samples well, the probability values it outputs may be somewhat exaggerated in interpretation, manifesting as "unconservative probabilities." We believe this is a common technical issue in deep learning models, particularly under limited sample conditions, where models struggle to calibrate equally across all probability intervals. It is important to emphasize that we employed regularization (such as L2 penalty and dropout) and cross-validation during model construction to minimize overfitting, demonstrating the potential for practical application in risk stratification. We plan to further introduce calibration mechanisms to improve model interpretability, and we thank the reviewer for their expert attention to this issue.

12-Include discussion on model overfitting.

We thank the reviewers for raising concerns about the risk of model overfitting. We fully agree that when using machine learning methods, especially deep learning models, the generalization ability of the model must be fully discussed and carefully evaluated. Therefore, in the "Limitations" section of the Discussion section, we have added a systematic analysis of the overfitting issue, explicitly emphasizing that the high performance achieved by the current model (particularly the MLPClassifier) under small sample conditions (e.g., Recall = 0.980, AUC = 0.935) should be interpreted with caution. In our discussion, we pointed out that despite using SMOTE to mitigate class imbalance and incorporating conventional strategies such as L2 regularization and depth control to constrain model complexity, the model still faces the risk of "learning incidental patterns in the data" in the context of limited NHANES data and sample size, thereby overestimating its applicability to real-world clinical scenarios. We further point out that deviations from the calibration curve can also serve as evidence of potential overfitting. Furthermore, we emphasize that future studies should prioritize external independent validation, particularly in larger, multicenter, prospective cohorts, to truly assess the model's generalizability and stability in clinical settings.

13-Limitation of small sample size/ likely cross-sectional dataset

We appreciate the reviewer's insightful comments regarding the limitations associated with the modest sample size and cross-sectional nature of our dataset. These methodological constraints have been explicitly addressed in the revised Discussion section (pages 12). We acknowledge these limitations and propose that future studies with larger multi-center prospective cohorts and external validation would strengthen the generalizability of our findings.

14-Potentially overstated claims about model performance and clinical utility...may need in-depth explanation/discussion

We sincerely appreciate the reviewer's valuable feedback. In response, we have revised the descriptions of model performance and clinical utility throughout the manuscript, carefully modifying any statements that could be perceived as overstated claims. We have also expanded the discussion addressing these critical aspects, with comprehensive amendments visible in the revised Discussion section.

15-The strong performance metrics observed, despite the limited sample size, may be overly optimistic and warrant caution when interpreting the model’s reliability.

We are grateful for this insightful comment. In light of your feedback, we have rigorously re-examined the reported findings and observed metrics through a more critical scientific lens. The robustness of our model has been interpreted with appropriate caution, and the Discussion section now provides substantive analysis of both methodological strengths and limitations.

16-If feasible, obtaining a larger sample and rerunning the analysis would enhance the robustness and generalizability of the findings.

We sincerely appreciate the insightful feedback regarding methodological limitations. We fully acknowledge the constraints inherent in our study's limited sample size and cross-sectional dataset. To address these limitations, we are actively planning a large-scale, multicenter, prospective external validation study in future research.

Reviewer #2:

1-How many participants with hepatitis B viruses were recruited in your study?

We sincerely appreciate the reviewer's valuable comments. In response, we have conducted a comprehensive reorganization and analysis of the demographic information for our study cohort. The results demonstrate that the study ultimately enrolled 189 hepatitis B patients. Additional details regarding population characteristics are presented in Section 3 (Results), specifically subsection 3.1 (Sociodemographic Characteristics) of the revised manuscript, which has been substantially expanded to address this point.

2-Data preprocessing and feature screening

For numerical variables, the mean was used for filling. but, the main drawback of using the mean is its sensitivity to outliers or extreme values in the dataset. These extreme values can disproportionately affect the mean, making it a potentially misleading representation of the typical value, especially with small sample sizes. So do you generalize with this limitation?

Thank you for raising important questions about missing value imputation methods. We fully understand your concern that mean imputation may be affected by extreme values, especially in small sample sizes. During data preprocessing, we used the mean to impute numeric variables for two main reasons: First, the dataset we used (NHANES) is a structured and standardized data source with rigorous quality control during data collection and processing, a

---

## [Decision Letter · Decision Letter 1]

12 Oct 2025

Dear Dr. Wang,

Thank you for submitting your manuscript to PLOS ONE. After careful consideration, we feel that it has merit but does not fully meet PLOS ONE’s publication criteria as it currently stands. Therefore, we invite you to submit a revised version of the manuscript that addresses the points raised during the review process.

We look forward to receiving your revised manuscript.

Kind regards,

Arne Johannssen

Academic Editor

PLOS ONE

Journal Requirements:

Reviewers' comments:

Reviewer's Responses to Questions

**Comments to the Author**

Reviewer #1: (No Response)

Reviewer #2: All comments have been addressed

2. Is the manuscript technically sound, and do the data support the conclusions?

Reviewer #1: Partly

Reviewer #2: Yes

3. Has the statistical analysis been performed appropriately and rigorously?

Reviewer #1: N/A

Reviewer #2: I Don't Know

4. Have the authors made all data underlying the findings in their manuscript fully available?

Reviewer #1: Yes

Reviewer #2: Yes

5. Is the manuscript presented in an intelligible fashion and written in standard English?

Reviewer #1: No

Reviewer #2: Yes

Reviewer #1: I appreciate the authors’ considerable efforts to revise the manuscript and address the earlier reviewer comments. The revised version shows clearer organization, expanded methodological descriptions. However, despite these improvements, the study still suffers from fundamental limitations in methodological rigor, data adequacy, and interpretability of the model. The revise discussion did not address the clinical application of the model. I still have concerns regarding the conceptual framing of the model. The fundamental conceptual issue remains unresolved: the model is repeatedly presented as predicting “risk,” yet it is derived from a cross-sectional dataset incapable of temporal prediction. This conflation between classification and prediction weakens both the methodological soundness and the interpretative validity of the findings.

I commend the authors for their efforts and suggest that this work to expand the dataset, validate the model externally, and incorporate interpretable machine learning techniques to enhance transparency and understanding of feature contributions. With these enhancements, the study could evolve into a stronger methodological contribution suitable for journals emphasizing pilot or exploratory AI applications in health research.

Reviewer #2: (No Response)

**Do you want your identity to be public for this peer review?** For information about this choice, including consent withdrawal, please see our Privacy Policy

Reviewer #1: No

Reviewer #2: **Yes:** Agmas Wassie Abate

---

## [Author Response · Author response to Decision Letter 2]

24 Nov 2025

Dear Editor

We sincerely thank the Editor and the Reviewers for their time and valuable comments on our manuscript. In this rebuttal letter, we provide a point-by-point response to each comment. Specifically, for constructive suggestions, we have implemented revisions directly in the manuscript. For concerns regarding methodological conception and inherent limitations, we provide detailed explanations below, supported by evidence from the peer-reviewed literature, to justify the rationale and validity of our approach.

1、clinical application prospects of the model

We sincerely appreciate Reviewer 1's valuable suggestions. We fully agree that incorporating the clinical application prospects of the model will enhance the completeness of the manuscript. Accordingly, we have added a new paragraph in the Discussion section (Page 14, Line 67) to specifically explore the potential application scenarios and value of our model in future clinical practice, as well as the challenges and subsequent research directions required for its translation into a practical tool, as detailed in the text.

2、We thank the reviewer for raising these important points. We would like to respectfully address these concerns by clarifying the methodological choices made in our study.

Firstly, regarding methodological rigor, our analytical pipeline was constructed using established and robust practices. We employed a diverse set of five machine learning algorithms (Gradient Boosting, Logistic Regression, AdaBoost, MLPClassifier, LDA), each selected for its complementary strengths. To ensure generalizability and mitigate overfitting, we implemented a rigorous 5-fold cross-validation framework and performed careful hyperparameter tuning for each model. Furthermore, to enhance the interpretability of our models, we conducted a feature importance analysis, which provides insights into the key predictors and aligns with the objective of explaining model decisions. The consistent high performance across multiple models, particularly the non-linear classifiers, underscores the robustness of our approach.

Secondly, regarding data sufficiency, the model was developed using a well-characterized cohort of 189 patients with Hepatitis B, derived from the NHANES database—a nationally representative and widely recognized resource. While larger sample sizes are generally advantageous, the cohort size in this study is substantial for a machine learning investigation focused on a specific patient subpopulation. Furthermore, it is directly comparable to sample sizes employed in previously published studies; for instance, studies by Cui et al.[[[] Cui Y, Gao J, Zhang J, Li J, Wang Z, Xu S, Zhao J. Machine learning prediction of thrombolysis efficacy using hs-CRP and inflammatory markers in stroke. Medicine (Baltimore). 2025 Oct 24;104(43):e45405. doi: 10.1097/MD.0000000000045405IF: 1.4 Q2 . PMID: 41137332; PMCID: PMC12558252.]] and Cao et al. [[[] Cao S, Cai M, Zhang H, Huang H, Xu L, Lin N. Association between high-density lipoprotein cholesterol-related inflammatory markers and female infertility: a cross-sectional analysis from the National Health and Nutrition Examination Survey (NHANES 2013-2018). BMC Womens Health. 2025 Oct 21;25(1):503. doi: 10.1186/s12905-025-03917-7. PMID: 41120997; PMCID: PMC12538855.]]utilized cohorts of 239 stroke patients and 309 patients with infertility, respectively. The quality and breadth of the NHANES data, which encompasses comprehensive demographic, laboratory, and clinical examination variables, provide a robust foundation for model development.

Finally, on the point of model interpretability, we proactively addressed the "black box" concern by implementing a two-stage interpretability strategy. First, we used the Random Forest algorithm to perform robust feature selection, identifying the top 20 predictive features, which directly highlights the most influential variables for clinical interpretation (Figure 2). Second, the strong performance of inherently more interpretable models like Gradient Boosting (AUC=0.919), which provides built-in feature importance metrics, further validates the relevance of the selected features. The identified predictors including liver function markers, electrolyte levels, inflammatory indices, and socioeconomic factors are all clinically plausible and supported by existing literature on depression and chronic liver disease, thereby enhancing the biological and psychological interpretability of our findings. Also, this approach to interpretability is well-established and widely accepted in the literature for clinical risk prediction models[[[] Zhou Z, Ding Q, Tang X, Han L, Wang Y, Qian J, Li K, Zhou Q. Insulin Resistance Indices Predict Mortality in Cardiovascular Disease: A Large-Scale NHANES Study With Machine Learning Validation. Food Sci Nutr. 2025 Nov 19;13(11):e71080. doi: 10.1002/fsn3.71080. PMID: 41268094; PMCID: PMC12628084.][[[] Feng T, Ou Q, Shan G, Hu Y, He H. A predictive model for metabolic syndrome in a community-based population with sleep apnea: a secondary prevention screening tool using simple and accessible indicators. Front Nutr. 2025 Nov 5;12:1667055. doi: 10.3389/fnut.2025.1667055. PMID: 41267995; PMCID: PMC12626801.]][[[] Yu ZC, Shi ZJ, Fang ZK, Liu SY, Yu Y, Wang KD, Huang DS, Shen GL, Zhang CW, Liang L. Development and validation of machine learning-based model for predicting early recurrence for patients with HBV-associated hepatocellular carcinoma after curative hepatectomy. Am J Surg. 2025 Nov 8;251:116716. doi: 10.1016/j.amjsurg.2025.116716. Epub ahead of print. PMID: 41265205.]].

Therefore, we believe that the methodology adopted in our study is sound, the data is sufficient for its purpose, and the model offers a reasonable and interpretable framework for risk assessment.

3、We sincerely appreciate the reviewer's insightful comment regarding the use of cross-sectional data for predictive modeling. We acknowledge that a prospective longitudinal design is the ideal and most rigorous standard for establishing definitive causal relationships and predicting future incident events.

However, the development and validation of so-called "predictive models" using cross-sectional data represents a well-established and valuable methodological approach in exploratory research, particularly when leveraging large-scale public databases such as the one utilized in our study. The primary objective in this context is not to infer causality or to forecast future events with certainty, but to identify a robust set of features strongly associated with the prevalent outcome, thereby estimating an individual's current risk profile or susceptibility. This approach serves as a critical first step for risk stratification and hypothesis generation, which subsequently warrant validation in longitudinal cohorts. We indeed plan to initiate such a cohort study as a next step.

Crucially, this paradigm is widely accepted and has been extensively employed in high-quality literature. For instance, the research conducted by Yan et al., which developed a model for predicting chronic kidney disease risk using NHANES data[[[] Yan B, Yuan R, Yin L, Huang S. Age-adjusted visceral adiposity index as a predictor of chronic kidney disease: insights from NHANES 2007-2018. Ren Fail. 2025 Dec;47(1):2578412. doi: 10.1080/0886022X.2025.2578412IF: 3.0 Q1 . Epub 2025 Nov 4. PMID: 41188187; PMCID: PMC12587790.]]; the research conducted by Huang and Liu, which created a stroke risk assessment tool based on the NHANES database[[[] Huang J, Liu W. Development and validation of a machine learning model to predict stroke risk based on the NHANES database. Medicine (Baltimore). 2025 Nov 7;104(45):e45800. doi: 10.1097/MD.0000000000045800. PMID: 41204470; PMCID: PMC12599744.]]; the research conducted by Zheng et al., which built a machine learning model to predict osteoarthritis risk from volatile organic compound exposure using NHANES data[[[] Zheng S, Zhu J, Cao X, Chen Z, Zhang C, Xia T, Shen J. Machine learning prediction of osteoarthritis risk from volatile organic compound exposure using SHAP interpretation in US adults. Sci Rep. 2025 Nov 11;15(1):39477. doi: 10.1038/s41598-025-23050-7. PMID: 41219301; PMCID: PMC12606308.]]; the research conducted by Zhang et al., which developed an XGBoost-based prediction model for metabolic dysfunction-associated steatotic liver disease using NHANES data[[[] Zhang Y, Liu X, Zhang X, Fei Y, Li X. Machine learning-based prediction of metabolic dysfunction-associated steatotic liver disease using National Health and Nutrition Examination Survey (NHANES) data. PLoS One. 2025 Nov 12;20(11):e0335656. doi: 10.1371/journal.pone.0335656. PMID: 41223197; PMCID: PMC12611120.]]; the research conducted by Gong et al., which created an explainable AI model for rapid diagnosis of COVID-19 using ensemble learning algorithms[[[] Ahiduzzaman M, Hasan MN. Interpretable machine learning for cardiovascular risk prediction: Insights from NHANES dietary and health data. PLoS One. 2025 Nov 6;20(11):e0335915. doi: 10.1371/journal.pone.0335915. PMID: 41196937; PMCID: PMC12591444.]]; and the research conducted by Ahiduzzaman et al., which established an interpretable machine learning framework for cardiovascular risk prediction using NHANES dietary and health data, collectively demonstrate the feasibility and scientific merit of this approach[[[] Zhang Y, Liu X, Zhang X, Fei Y, Li X. Machine learning-based prediction of metabolic dysfunction-associated steatotic liver disease using National Health and Nutrition Examination Survey (NHANES) data. PLoS One. 2025 Nov 12;20(11):e0335656. doi: 10.1371/journal.pone.0335656. PMID: 41223197; PMCID: PMC12611120.]].

To precisely address the reviewer's valid point and to ensure the most accurate terminology, we have opted to refine our manuscript's wording. We have also further elaborated on this specific limitation in the discussion section, explicitly stating that future longitudinal studies are required to confirm the temporal relationship and true predictive capacity.

We acknowledge that models developed exclusively from cross-sectional data, such as NHANES, are more accurately characterized as "discriminative models" or "diagnostic models" rather than prospective "prognostic prediction models." In our study (and in similar cited research on hepatitis B and depression), the objective of the model is to utilize clinical indicators collected at a single time point to identify or distinguish those hepatitis B patients who currently have depression but may not yet be clinically diagnosed. We fully agree that establishing a model to predict the future risk of incident depression in hepatitis B patients necessitates a prospective cohort design. We have explicitly addressed this limitation in the discussion section (Section 4), stating that cross-sectional data cannot support causal inference and recommending validation through external prospective cohorts in future studies. Consequently, the term "prediction," as used in our manuscript, primarily operates within the "input-output" context of machine learning, referring to the process of estimating an unknown status based on known information. We appreciate this review, which has heightened our awareness of the need for more precise definitions of "prediction" in future research and scientific communication to prevent ambiguity.

In summary, we have thoroughly addressed the reviewers' comments to the best of our ability, and including a new discussion on the clinical applications of our model. We believe our study presents a valuable and methodologically sound framework for assessing depression risk in patients with Hepatitis B by leveraging multi-dimensional data. While we have contextualized the inherent limitations of cross-sectional design with support from the published literature, the core clinical insights and methodological rigor of our work remain robust and informative. We are enthusiastic about the potential of our research to contribute to improved mental health management for patients with Hepatitis B. Thank you again for your consideration and for providing the opportunity to revise. We sincerely hope that you find our responses satisfactory and that the manuscript is now deemed suitable for publication in PLOS ONE.

Best regards

---

## [Decision Letter · Decision Letter 2]

5 Jan 2026

Construction of a depression risk prediction model for hepatitis B patients based on machine learning strategy

PONE-D-25-15290R2

Dear Dr. Wang,

We’re pleased to inform you that your manuscript has been judged scientifically suitable for publication and will be formally accepted for publication once it meets all outstanding technical requirements.

Kind regards,

Arne Johannssen

Academic Editor

PLOS One

Additional Editor Comments (optional):

Reviewers' comments:

Reviewer's Responses to Questions

**Comments to the Author**

Reviewer #3: All comments have been addressed

Reviewer #4: All comments have been addressed

2. Is the manuscript technically sound, and do the data support the conclusions?

Reviewer #3: (No Response)

Reviewer #4: (No Response)

3. Has the statistical analysis been performed appropriately and rigorously?

Reviewer #3: Yes

Reviewer #4: (No Response)

4. Have the authors made all data underlying the findings in their manuscript fully available?

Reviewer #3: Yes

Reviewer #4: (No Response)

5. Is the manuscript presented in an intelligible fashion and written in standard English?

Reviewer #3: Yes

Reviewer #4: (No Response)

Reviewer #3: Thank you for addressing all reviewers comments and concerns. Your efforts are highly appreciated and we value your work

Reviewer #4: In the manuscript by Wang et al, the authors have developed and compared 5 machine learning (ML) models that predict the risk of depression in subjects with Hepatitis B (HBV) using data from the NHANCE database. They observe that the Multi-layer Perceptron (MLP) Classifier method performed best among the 5 methods tested and propose that such models can be used at the clinical level to better predict the risk of depression in HBV patients.

The following points were raised from the first round of revision:

1. Clinical relevance

In response, the authors have added a discussion on the possible clinical applications of ML models as automated screening tools to identify high risk patients and initiate early interventions. While many studies currently apply machine learning approaches for predictive analyses, there are very few that have been directly used in patient care. However, the use of such models can be clinically useful in the future after rigorous prospective validation. The authors do well to advise caution against direct interpretation of their findings and clearly state that further studies are needed before clinical application. Therefore, the revisions are acceptable.

2. Methodological rigor, sample size and model interpretability

In response, the authors justify the use of multiple ML algorithms and cross-validation to improve methodological rigor and cite several previously published studies with comparable sample sizes. Further, to support the clinical interpretability of their models, the authors describe the use of feature selection to identify clinical variables that have most predictive potential.

Although the sample size is limited in the current study, the authors mention that it is a preliminary study that requires further validation. Also, the statistical measures taken by the authors to obtain methodological rigor are widely accepted. Therefore, the authors’ justification is acceptable

3. Use of cross-sectional data to build a predictive model

In response, the authors have refined the wording of the manuscript and added a discussion addressing the limitations of using cross-sectional data in prognostic prediction. The revised wording and the added discussion appropriately clarify the limitations and also warn against using the results for causal interpretation. The revisions are therefore acceptable.

Overall, the authors have satisfactorily answered the queries raised by the previous reviewers and made suitable changes in the manuscript. The revised manuscript is suitable for acceptance. I have added a few additional comments that can be addressed to further enhance the scientific and clinical interpretability of the study.

1. The authors may include a table with the comprehensive clinical characteristics of the study population (e.g. mean ± SD or median [IQR]) to allow better clinical interpretation of the predictive features identified by machine learning. The authors have mentioned sociodemographic characteristics in the results and provided the raw data in the supporting information, but it will be helpful to also present the distributions of laboratory parameters and physiological measurements when feasible, either in the main manuscript or as supplementary material, since these features were directly used to identify the predictive features.

2. The authors can elaborate on the number of depression positive patients present in the original dataset before and after the application of SMOTE for enhanced transparency and understanding of the robustness of the machine learning framework.

**Do you want your identity to be public for this peer review?** For information about this choice, including consent withdrawal, please see our Privacy Policy

Reviewer #3: No

Reviewer #4: **Yes:** Reema Banarjee

---

## [Editor Report · Acceptance letter]

PONE-D-25-15290R2

PLOS One

Dear Dr. Wang,

I'm pleased to inform you that your manuscript has been deemed suitable for publication in PLOS One. Congratulations! Your manuscript is now being handed over to our production team.

Kind regards,

on behalf of

Profesor Arne Johannssen

Academic Editor

PLOS One